# A Multimodal Approach for Clinical Diagnosis and Treatment of Primary Progressive Aphasia (MAINSTREAM): A Study Protocol

**DOI:** 10.3390/brainsci13071060

**Published:** 2023-07-12

**Authors:** Maria Cotelli, Francesca Baglio, Rosa Manenti, Valeria Blasi, Daniela Galimberti, Elena Gobbi, Ilaria Pagnoni, Federica Rossetto, Emanuela Rotondo, Valentina Esposito, Roberto De Icco, Carla Giudice, Cristina Tassorelli, Eleonora Catricalà, Giulia Perini, Cristina Alaimo, Elena Campana, Luisa Benussi, Roberta Ghidoni, Giuliano Binetti, Tiziana Carandini, Stefano Francesco Cappa

**Affiliations:** 1Neuropsychology Unit, IRCCS Istituto Centro San Giovanni di Dio Fatebenefratelli, 25125 Brescia, Italy; mcotelli@fatebenefratelli.eu (M.C.); rmanenti@fatebenefratelli.eu (R.M.); egobbi@fatebenefratelli.eu (E.G.); ipagnoni@fatebenefratelli.eu (I.P.); calaimo@fatebenefratelli.eu (C.A.); ecampana@fatebenefratelli.eu (E.C.); 2IRCCS Fondazione Don Carlo Gnocchi ONLUS, 20148 Milan, Italy; vblasi@dongnocchi.it (V.B.); frossetto@dongnocchi.it (F.R.); 3Fondazione IRCCS Ca’ Granda Ospedale Maggiore Policlinico, 20122 Milan, Italy; daniela.galimberti@unimi.it (D.G.); emanuela.rotondo@gmail.com (E.R.);; 4Deparment of Biomedical, Surgical and Dental Sciences, University of Milan, 20122 Milan, Italy; 5Dementia Research Center, IRCCS Mondino Foundation, 27100 Pavia, Italy; valentina.esposito@mondino.it (V.E.); giulia.perini@mondino.it (G.P.); stefano.cappa@iusspavia.it (S.F.C.); 6Department of Brain and Behavioral Sciences, University of Pavia, 27100 Pavia, Italy; roberto.deicco@mondino.it (R.D.I.); carla.giudice@mondino.it (C.G.); cristina.tassorelli@mondino.it (C.T.); 7Movement Analysis Research Unit, IRCCS Mondino Foundation, 27100 Pavia, Italy; 8ICoN Cognitive Neuroscience Center, Institute for Advanced Studies, IUSS, 27100 Pavia, Italy; eleonora.catricala@iusspavia.it; 9Molecular Markers Laboratory, IRCCS Istituto Centro San Giovanni di Dio Fatebenefratelli, 25125 Brescia, Italy; lbenussi@fatebenefratelli.eu (L.B.); rghidoni@fatebenefratelli.eu (R.G.); 10MAC-Memory Clinic and Molecular Markers Laboratory, IRCCS Istituto Centro San Giovanni di Dio Fatebenefratelli, 25125 Brescia, Italy; gbinetti@fatebenefratelli.eu

**Keywords:** primary progressive aphasia, language training, transcranial Direct Current Stimulation, imaging biomarkers, molecular biomarkers

## Abstract

Primary Progressive Aphasia (PPA) is a syndrome due to different neurodegenerative disorders selectively disrupting language functions. PPA specialist care is underdeveloped. There are very few specialists (neurologists, psychiatrists, neuropsychologists, and speech therapists) and few hospital- or community-based services dedicated to the diagnosis and continuing care of people with PPA. Currently, healthcare systems struggle to provide adequate coverage of care that is too often fragmented, uncoordinated, and unresponsive to the needs of people with PPA and their families. Recently, attention has been gained by non-invasive brain stimulation techniques that allow a personalized treatment approach, such as transcranial Direct Current Stimulation (tDCS). The MAINSTREAM trial looks forward to introducing and evaluating therapeutic innovations such as tDCS coupled with language therapy in rehabilitation settings. A Multimodal Approach for Clinical Diagnosis and Treatment of Primary Progressive Aphasia, MAINSTREAM (ID: 3430931) was registered in the clinicaltrials.gov database (identifier: NCT05730023) on 15 February 2023.

## 1. Introduction

The Global Burden of Disease (GBD) study on disability revealed that 2.41 billion individuals worldwide had conditions that would benefit from rehabilitation and that this need was essentially unmet [1]. This is particularly true for neurological disorders, which are a leading cause of death (nearly 10 million deaths in 2019) and disability (349 million disability-adjusted life-years (DALYs) in the same period). Alzheimer’s and other dementias are one of the four leading contributors to neurological disability, and the number of DALYs continues to increase with an aging population [2]. The scarcity of established data on the determinants and outcomes of neurodegenerative conditions indicates that new knowledge is required to develop effective prevention and treatment strategies. Moreover, the demand for pharmacological and non-pharmacological treatments is increasing and has essential health service implications for managing these conditions. In this field, the Multimodal Approach for Clinical Diagnosis and Treatment of Primary Progressive Aphasia (MAINSTREAM) trial focuses on Primary Progressive Aphasia (PPA). PPA is a neurological syndrome due to different neurodegenerative disorders that selectively and progressively disrupt language functions [3], with a progressive impact on patients’ relationships, social networks, and participation in everyday activities based on communication [4]. PPA is characterized by various clinical symptoms of speech and language impairment due to the degeneration of language networks with heterogeneous degrees of severity [5]. The specificity of the clinical symptoms, reflecting the involved networks, enables the identification of different PPA phenotypes. The guidelines for the classification of PPAs have revealed three main variants: the non-fluent/agrammatic variant of PPA (nf/avPPA), the semantic variant of PPA (svPPA), and the logopenic/phonological variant of PPA (l/phvPPA) [6,7]. Moreover, the consensus criteria acknowledge that some patients do not fulfil the criteria for any PPA variants and are defined as “PPA unclassifiable” [7,8]. Each clinical subtype is characterized by the involvement of different linguistic areas and circuits: in nf/avPPA, degeneration involves the insular cortex, the posterior–inferior frontal gyrus (Broca’s region) and the superior temporal gyrus; in l/phvPPA, the progressive atrophy mostly affects the temporal–parietal junction; and in svPPA, atrophy is most profound in the anterior temporal lobes, including their mesial and inferior portions. Moreover, the PPA variants are probabilistically associated with different neuropathological substrates: frontotemporal lobar degeneration (FTLD) is associated with transactive response DNA-binding proteinopathy (TDP), FTLD-tau (nf/avPPA) or with Alzheimer’s disease (AD) pathology (l/phvPPA). As summarized, the knowledge about the neural substrates and mechanisms of PPA is limited. Currently, early diagnosis markers are unavailable, which might otherwise facilitate access to timely and effective therapies for persons living with PPA (plPPA).

### 1.1. Prior Research (Literature Review)

Interest in non-pharmacological therapies such as language training interventions, aimed at reducing the functional impact of communication difficulties on daily life and remediate language deficits in PPAs, has increased remarkably in the last decades [3,9]. An important issue is the application of a tailored approach based on language deficits and the patient’s needs [10] to be pursued in the early stages of the disease [11,12]. The literature reports different therapeutic approaches that have been utilized to create language treatments tailored to the individual PPA patient [13,14,15,16,17,18,19,20,21,22,23,24,25], inducing both immediate and long-term effects (e.g., [24,26]). However, most studies investigated the effectiveness of naming treatments in patients with a progressive word-finding difficulty [10,11].

We conducted a literature review and meta-analysis on the language treatments most commonly used in clinical practice among patients with different variants of PPA, focusing on enhancing spoken and written naming abilities [12,27]. Generally, the review indicated that language training could induce immediate post-treatment improvements in naming abilities in all variants of PPA. Moreover, despite the large variability in the results, the generalization and long-term effects of treatment gains may be noted after the training. Specifically, one well-investigated form of language intervention for plPPA is the lexical retrieval treatment [13]. Several treatment techniques target lexical retrieval [11,28]; for this reason, this approach is used flexibly in all variants of PPA. Phonological and orthographic treatment is most frequently dedicated to patients with a diagnosis of l/phvPPA, while semantic treatment is mainly applied to svPPA patients [12]. 

Some studies have highlighted the positive effects of telerehabilitation-based language treatment in PPA [22,29]. As many plPPA have limited access to traditional treatments, the use of telerehabilitation-based language treatment in PPA could help deliver care in a more widespread fashion [29]. Notably, Dial, et al. Dial, et al. [29] administered a treatment protocol aimed at promoting speech production and fluency in svPPA, l/phvPPA and nf/avPPA patients via either face-to-face or telerehabilitation appointments. Interestingly, no significant differences were found in terms of benefit, regardless of the modality of the intervention’s administration (face-to-face vs. telerehabilitation). This finding supports the application of telerehabilitation as an innovative instrument for delivering treatment to plPPA.

In recent years, there has also been a growing interest in the use of non-invasive stimulation techniques, such as transcranial Direct Current Stimulation (tDCS) and repetitive transcranial magnetic stimulation (rTMS) in PPA patients. Overall, treatment with rTMS or tDCS in plPPA yielded improved language abilities with respect to verbal fluency, written abilities, oral production, repetition, reading, comprehension, and naming after the stimulation [30]. Interestingly, some studies have suggested that using non-invasive brain stimulation techniques combined with language intervention increases the benefits of language training in plPPA [3,27,30,31]. Tippett, et al. Tippett, et al. [3] showed that language intervention improves language outcomes in plPPA, and tDCS can enhance the generalization and maintenance of positive language outcomes. A recent systematic review and meta-analysis from our group investigated the efficacy of language training on naming abilities in PPA, either alone or in combination with non-invasive brain stimulation [27]. It was shown that only language training combined with tDCS could improve the naming accuracy for untrained items, thus demonstrating a generalization effect. Moreover, the advantages of tDCS were evident at follow-up visits [27]. Finally, a randomized, double-blind, sham-controlled study applied tDCS associated with written naming/spelling therapy, revealing that tDCS can increase the effectiveness of language therapy in plPPA [32]. In addition, the study showed that tDCS efficacy differed across PPA variants, with greater effects of the combined treatment (tDCS plus spelling therapy) on written naming abilities in nf/avPPA and l/phvPPA. In contrast, in the svPPA patients, tDCS did not significantly improve the effectiveness of language therapy for trained and untrained items [32]. These studies suggest that the use of these non-invasive techniques coupled with language intervention is more effective than language training or non-invasive brain stimulation applied alone [27,30]. 

Despite the aforementioned benefits of treatment, the neural and biological correlates of cognitive improvements induced by rehabilitation techniques in PPA and the variables that might be related to successful recovery remain poorly understood [33]. Moreover, the prediction of individual patient responses to combined behavioral and neuromodulation treatment remains an elusive goal. 

### 1.2. Objectives

In this double-blind, randomized, controlled pilot study, we aim to test tDCS neuromodulation coupled with language therapy in rehabilitation settings. In particular, we will evaluate the short-term and long-term effects of active tDCS over the dorsolateral prefrontal cortex (DLPFC) combined with individualized language training vs. placebo DLPFC-tDCS with individualized language training in a large sample of patients with mild PPA. In particular, we aim to establish if the combined use of active tDCS and individualized language training in mild PPA patients can improve cognitive and functional performance. Moreover, we aim to evaluate the changes in brain activity and molecular markers as they relate to the intervention. To date, little is known about the mechanisms of the treatment’s effects. This trial could partially fulfil the need for new rehabilitation instruments to be used in daily disease management. It will increase understanding of the neural and biological characteristics linked to the effect of language treatment and non-invasive brain stimulation application in PPA.

## 2. Materials and Methods

The protocol of the present study has been prepared as outlined in the “Standard Protocol Items: Recommendations for Interventional Trials” (SPIRIT) guidelines (Figure 1).

The study will be conducted in accordance with the Declaration of Helsinki. It has been approved by the Ethics Committees of the Istituto di Ricovero e Cura a Carattere Scientifico (IRCCS) Centro San Giovanni di Dio, Fatebenefratelli, Brescia, Italy (protocol number: 52-2022), of the IRCCS Fondazione Mondino, Pavia, of the Fondazione IRCCS Ca’ Granda, Ospedale Maggiore Policlinico, Milan and of IRCCS Fondazione Don Carlo Gnocchi, Milan.

### 2.1. Trial Design and Setting

This multicentric study is a double-blinded, randomized, controlled pilot study involving 60 patients with mild PPA, defined using the Progressive Aphasia Severity Scale (PASS) [34,35], recruited from IRCCS Istituto Centro San Giovanni di Dio, Fatebenefratelli, Brescia; IRCCS Fondazione Mondino, Pavia; Fondazione IRCCS Ca’ Granda Ospedale Maggiore Policlinico, Milan, and Fondazione Don Carlo Gnocchi—ONLUS, Milan, Italy.

After recruitment, the subjects will be randomized into one of two parallel groups: (i) active tDCS over the DLPFC combined with an individualized language training group or (ii) placebo DLPFC-tDCS with an individualized language training group. The two groups will be matched for age, education, and language deficit severity. All participants will undergo a clinical, cognitive, and language evaluation at the baseline (T0), post-treatment (T1), and at the 3-month (T2) follow-up. Magnetic Resonance Imaging (MRI) and biomolecular data will be collected at T0 and T1 (after the treatment). The trial work plan is shown in Figure 2.

### 2.2. Sample Size

The sample size calculation was computed using G*Power 3 software [36,37]. The sample size for this pilot study was determined based on previous work [38] and was computed for the primary outcome variable (an increase in naming accuracy). An eta square of 0.25 of the interaction between time and the experimental group was considered for the effect size calculation. A minimum sample size of 12 subjects is required for a 90% powered study (alpha = 0.05). The resulting number was increased to 30 subjects per group based on the 20% expected dropout rate among the subjects and because of the considerable individual variability expected from the inclusion of all PPA variants.

### 2.3. Study Population, Recruitment, and Randomization

According to the sample size calculation, the MAINSTREAM trial has a target enrollment of 60 patients diagnosed with mild PPA, defined using the Progressive Aphasia Severity Scale (PASS) [34,35]. Eligible patients who meet all inclusion criteria (see the paragraph below) will be randomized using an adaptive randomization procedure. Clinicians and patients will be blinded to the trial intervention. The group allocation will be masked to the statistician conducting the data analysis.

### 2.4. Inclusion and Exclusion Criteria

Formal criteria-based diagnosis of PPA will be applied, and the variant will be determined according to consensus criteria [7].

The inclusion criteria for participants will be:
Age eligible for the study: 40 years or older (Adult, Older Adult); Agreement to participate by signing the informed consent form; Diagnosis of PPA according to the current clinical criteria [7]; Mild PPA defined using the Progressive Aphasia Severity Scale (PASS) [34,35];Native Italian speakers.

The exclusion criteria will be:Presence of developmental disorders;Presence of any medical or psychiatric illness that could interfere with completing assessments;Presence of any medical condition representing a contraindication to tDCS.

### 2.5. Trial Interventions

All patients will undergo treatment sessions of 45 min each on five consecutive days a week for two weeks: -30 patients will receive active tDCS over DLPFC (anode over the left DLPFC with the cathode over the right supraorbital region) while performing individualized language training;-30 patients will receive placebo tDCS over DLPFC during individualized language training.

#### 2.5.1. tDCS

All patients will receive two weeks of tDCS stimulation (Active or Placebo) during the individualized language training. Active tDCS (anode over the left DLPFC with the cathode over the right supraorbital region) will be delivered by a battery-driven stimulator (HDCStim System, Newronika) through a pair of saline-soaked sponge electrodes (35 cm^2^ each). The active electrode will be applied over the left DLPFC, and the reference electrode will be fixed over the right supraorbital region. We will develop a model based on the patient’s MRI average data to find the optimal electrode montage since cortical thinning and ventricular enlargement could substantially affect the current pathway. A constant current of 2 mA will be applied for 25 min (current density of active electrode 0.06 mA/cm^2^) with a ramping period of 10 s at the beginning and end of the stimulation, starting five minutes before the beginning of the language training. The current density will be maintained below the safety limits [39,40]. For placebo stimulation, the setting will be the same, but the current will be turned off shortly after initiation.

#### 2.5.2. Language Training

The language treatment will be based on the combination of compensation approaches and individualized language training, including an impairment-directed treatment approach for naming. An individual trained and untrained pictures list will be selected as detailed in the “*Primary outcome measures*” paragraph below. The subjects will receive individualized language training treatments that include several steps designed to encourage the strategic recruitment of spared semantic, orthographic, and phonological knowledge as well as episodic/autobiographical information to facilitate word retrieval and to elicit the production of the target object [26,32,41]. In this regard, the individualized language training will incorporate elements of the most frequently used treatments in clinical practice among patients with different variants of PPA, such as lexical retrieval treatment [11]; phonological and/or orthographic treatment [42,43,44]; and semantic treatment [17,45]. At each step, in the case of difficulties or errors, the patient will be helped by the therapist to provide the correct response. The patient will be seated in front of a computer screen in a quiet room while the therapy protocol will be displayed using Microsoft PowerPoint. The pictures will show the items to be trained, and the treatment will involve several steps:

*Step 1: Semantic Feature Analysis*. The patient will be asked questions that are semantically related to the image presented in the center of the computer screen *(e.g., Is this a?; Where do you find it?; What does it look like?/What is it made of?/What color is it?; It reminds me of?; What is it for?/What is it used for?/Which verb can be associated?*). In the case of difficulty in answering any question, the therapist will make suggestions.

*Step 2: Script Training*. The patient will be asked to formulate a sentence using the verb evoked during the previous step (Semantic Feature Analysis). Subsequently, the patient will be asked to read aloud the sentence presented on the screen; in the case of reading difficulties, the sentence will be produced by the therapist, and the subject will be asked to repeat it. 

*Step 3: Phonemic cue*. The participant will be asked to orally produce the initial sound of the word corresponding to the image presented in the center of the screen. In the case of an error, the initial letter will be shown, and the patient will be asked to produce the sound. For the next cue, the correct sound will be pronounced by the therapist, and the participant will be asked to repeat it. 

*Step 4: Oral reading*. The target written word will be presented on the computer screen, and the participant will read it aloud; in the case of reading difficulties, the word will be produced by the therapist, and the subject will be asked to repeat it.

*Step 5: Articulatory suppression task*. This involves interference with articulatory codes caused by the uttering of an irrelevant speech sound (i.e., bla, bla, bla). In the suppression condition, participants will receive instructions to start uttering the syllable “bla”. 

*Step 6: Oral picture naming*. The target picture will be presented on the computer screen, and the participant will be asked to retrieve its correct name.

*Step 7: Oral repetition*. The target word will be spoken by the therapist, and the participant will be asked to repeat it three times. 

*Step 8: Articulatory suppression task*. This involves interference with articulatory codes caused by the uttering of an irrelevant speech sound (i.e., bla, bla, bla).

*Step 9: Oral picture naming*. The target picture will be presented on the computer screen, and the participant will be asked to retrieve its correct name.

At the end of this treatment, the therapist will have 20 additional minutes to focus on other treatment strategies based on the individual patient’s language difficulties. 

The rehabilitation program will include a therapist guide and a patient workbook. The therapist guide will walk the therapist through the program and will explain what to say and do with the patient in each session.

### 2.6. Outcome Measures

#### 2.6.1. Primary Outcome Measures

The primary outcome measure will be the change from baseline (T0) to post-treatment (T1) and follow-up assessment (T2) in object- and action-naming task scores, as assessed by the International Picture Naming Project (IPNP) Task [46]. 

At the baseline assessment (T0), the patients will undergo two sessions of oral object naming to select object stimuli for the treatment (trained items) and control object stimuli for the assessment of generalization effects (untrained items). Pictures for the oral naming task will include 300 black-and-white drawings of objects taken from the UCSD Center for Research in Language—International Picture-Naming Project corpus [46]. This database provides norms and lexical information (frequency, age of acquisition, etc.) for picture naming in seven languages. The pictures have been tested and normed on healthy and patient populations [46]. In this protocol, the pictures for the oral naming task will be displayed twice (on two consecutive days) on a computer screen using Microsoft PowerPoint; each picture will be presented for a maximum of 6 s. Each participant will be asked to name the object, and oral responses will be recorded.

The pictures not correctly named by the participant in at least one out of the two oral naming sessions will be further split into two sets: the “therapy” list, which will include the items to be trained (trained stimuli), and the control stimuli list, which will include items not to be trained (untrained stimuli). The two lists will be balanced as closely as possible for several variables (e.g., word frequency, number of syllables, semantic category, etc.). The procedure that will be applied to select the “therapy” and “control” lists will produce a personalized set of items for each participant, which ensures the within- and across-subject validity of the design. The accuracy in naming trained and untrained items will be assessed at the end of the rehabilitation (T1) and during follow-up visits (T2). 

Moreover, naming oral action items will be evaluated to assess generalization effects. 

#### 2.6.2. Secondary Outcome Measures

Secondary outcome measurements will be the changes from baseline (T0) to post-treatment (T1) and follow-up assessment (T2) in standardized clinical, communication, and functional abilities questionnaires, as well as neuropsychological tests for each cognitive domain and specific language task. 

In particular, we will compute changes in clinical, communication, and functional abilities: Stroke and Aphasia Quality of Life Scale—39 (SAQoL—39 [47]): measures the health-related quality of life in people with chronic aphasia. It provides a total score from 1 to 5 (higher score = better outcome), and it analyzes the following four domains: physical (score range: min = 1, max = 5; higher score = better outcome); psychosocial (score range: min = 1, max = 5; higher score = better outcome); communication (score range: min = 1, max = 5; higher score = better outcome), and energy (score range: min = 1, max = 5; higher score = better outcome);Communication Outcome After Stroke (COAST [48]): a measure of functional communication in daily activities and of its impact on the quality of life from the point of view of the aphasia patient (COAST total score range: min = 0, max = 80; higher score = better outcome) and their carer (Carer COAST total score range: min = 0, max = 80; higher score = better outcome);Functional Outcome Questionnaire-aphasia [49]: measures an aphasia patient’s functional and pragmatic communication in the home and community settings. It provides a total score from 32 to 160 (higher score = better outcome), and it is divided into four subscales: communicating basic needs (CBN score range: min = 1, max = 35; higher score = better outcome); making routine requests (MRR score range: min = 1, max = 35; higher score = better outcome); communicating new information (CNI score range: min = 1, max = 40; higher score = better outcome), and attention/other communication skills (AO score range: min = 1, max = 50; higher score = better outcome);Frontotemporal Dementia—Clinical Dementia Rating Scale (FTD-CDR)—Sum of Boxes [50]: a measure of dementia severity and progression in FTD (score range: min = 0, max = 24; higher score = worse outcome);Global Clinical Dementia Rating (CDR) plus NACC FTLD [51]: a measure of dementia severity and progression in FTD (score range: min = 0, max = 3; higher score = worse outcome);Progressive Aphasia Severity Scale (PASS [34,35]): measures the severity and progression of language deficits in patients with PPA. The scale assesses ten linguistic domains (each score range: min = 0, max = 3): articulation (higher score = worse outcome), fluency (higher score = worse outcome), syntax and grammar (higher score = worse outcome), word retrieval and expression (higher score = worse outcome), repetition (higher score = worse outcome), auditory comprehension (higher score = worse outcome), single-word comprehension (higher score = worse outcome), reading (higher score = worse outcome), writing (higher score = worse outcome), and functional communication (higher score = worse outcome). Moreover, it analyzes three supplemental domains (each score range: min = 0, max = 3): initiation of conversation (higher score = worse outcome), turn-taking during conversation (higher score = worse outcome), and generation of language (higher score = worse outcome);Lincoln Speech Questionnaire [52]: measures functional communication skills. It provides a speech score (score range: min = 0, max = 14; higher score = better outcome) and an understanding score (score range: min = 0, max = 5; higher score = better outcome);Communication Severity Rating Scale (Goodglass and Kaplan [53]): a measure of aphasia severity (score range: min = 0, max = 5; higher score = better outcome);Beck Depression Inventory (BDI [54]): a measure of depressive symptoms (score range: min = 0, max = 63; higher score = worse outcome);Frontal Behavioral Inventory (FBI [55]): a measure of behavior and personality (score range: min = 0, max = 72; higher score = worse outcome).

Moreover, the neuropsychological battery will include tests for the assessment of global cognition (Mini Mental State Examination, MMSE [56,57]; score range: min = 0, max = 30; higher score = better outcome) and for the deep evaluation of the following domains:

##### Language

Picture Naming subtest from Screening for Aphasia in NeuroDegeneration (SAND [58,59]): a measure of naming abilities. It provides a total score (score range: min = 0, max = 14; higher score = better outcome), a living score (score range: min = 0, max = 7; higher score = better outcome), and a non-living score (score range: min = 0, max = 7; higher score = better outcome);Auditory sentence comprehension subtest from Screening for Aphasia in NeuroDegeneration (SAND [58,59]): a measure of comprehension abilities (total score range: min = 0, max = 8; higher score = better outcome);Single-word comprehension subtest from Screening for Aphasia in NeuroDegeneration (SAND [58,59]): a measure of comprehension abilities. It provides a total score (score range: min = 0, max = 12; higher score = better outcome), a living score (score range: min = 0, max = 6; higher score = better outcome), and a non-living score (score range: min = 0, max = 6; higher score = better outcome);Repetition subtest from Screening for Aphasia in NeuroDegeneration (SAND [58,59]): a measure of repetition abilities. It provides a total score (score range: min = 0, max = 10; higher score = better outcome), a words score (score range: min = 0, max = 6; higher score = better outcome), and a non-words score (score range: min = 0, max = 4; higher score = better outcome);Sentence repetition subtest from Screening for Aphasia in NeuroDegeneration (SAND [58,59]): a measure of repetition abilities. It provides a total score (score range: min = 0, max = 6; higher score = better outcome), a predictable sentences score (score range: min = 0, max = 3; higher score = better outcome), and an unpredictable sentences score (score range: min = 0, max= 3, higher score = better outcome);Reading subtest from Screening for Aphasia in NeuroDegeneration (SAND [58,59]): a measure of reading abilities. It provides a total score (score range: min = 0, max = 16; higher score = better outcome), a words score (score range: min = 0, max= 12; higher score = better outcome), and a non-words score (score range: min = 0, max = 4; higher score = better outcome);Writing subtest from Screening for Aphasia in NeuroDegeneration (SAND [58,59]): a measure of writing abilities. It provides an Information Units score (score range: min = 0, max = 6; higher score = better outcome), a total number of words score (score range: min = 0, max = no limits; higher score = better outcome), a number of nouns/total number of words score (score range: min = 0, max = 1; higher score = better outcome), a number of verbs/total number of words (score range: min = 0, max = 1; higher score = better outcome), a number of correct syntactic structures/total number of syntactic structures score (score range: min = 0, max = 1; higher score = better outcome), a number of orthographic errors score (score range: min = 0, max = no limits; higher score = worse outcome), and a number of lexico-semantic errors/number of words score (score range: min = 0, max = 1; higher score = worse outcome);Semantic association subtest from Screening for Aphasia in NeuroDegeneration (SAND [58,59]): a measure of semantics (total score range: min = 0, max = 4; higher score = better outcome);Picture description subtest from Screening for Aphasia in NeuroDegeneration (SAND [58,59]): a measure of oral production. It provides an Information Units score (score range: min = 0, max = 8; higher score = better outcome), a total number of words score (score range: min = 0, max = no limits; higher score = better outcome), a number of nouns/total number of words score (score range: min = 0, max = 1; higher score = better outcome), a number of verbs/total number of words score (score range: min = 0, max = 1; higher score = better outcome), a number of sentences score (score range: min = 0, max = no limits; higher score = better outcome), a number of subordinates/number of sentences score (score range: min = 0, max = no limits; higher score = better outcome), a number of repaired sequences/number of words score (score range: min = 0, max = 1; higher score = worse outcome), a number of phonological errors/number of words (score range: min = 0, max = 1; higher score = worse outcome), and a number of lexico-semantic errors/number of words (score range: min = 0, max = 1; higher score = worse outcome);Naming Subtest from the Aachener Aphasia Test [60]: a measure of oral naming (score range: min = 0, max = 120; higher score = better outcome);Object naming subtest from Battery for the Assessment of Aphasic Disorders (BADA [61]): a measure of object naming (score range: min = 0, max = 30; higher score = better outcome);Verbal Fluency (semantic and phonemic [62]): a measure of verbal fluency abilities (score range min = 0, max = no limits; higher score = better outcome).

##### Memory

Story Recall [63]: a measure of verbal long-term memory (score range min = 0, max = 28; higher score = better outcome);Rey-Osterrieth Complex Figure—Recall [64]: a measure of nonverbal long-term memory (score range min = 0, max = 36; higher score = better outcome).

##### Attention and Executive Functions

Trial Making Test [65]: a measure of attentional abilities (score range: min = n/a, max = no limits; higher score = worse outcome);Stroop Test [66]: a measure of executive abilities (score range: min = n/a, max = no limits; higher score = worse outcome).

##### Constructional Praxis

Rey-Osterrieth Complex Figure—Copy [64]: a measure of constructional praxis (score range min = 0, max = 36; higher score = better outcome).

#### 2.6.3. Surrogate Outcome Markers

Surrogate outcome measures will be the changes from baseline (T0) to post-treatment (T1) in molecular biomarkers and structural and functional connectivity. 

In plasma, we will analyze the following: The size and the concentration of plasma extracellular vesicles (EVs) by nanoparticle tracking analysis, employing the Nano-Sigh Instrument (Malvern, Worcestershire, United Kingdom);Neurofilament light chain (NFL) levels and Glial Fibrillary Acidic Protein (GFAP) levels using the commercially available Neurology 2-Plex B array (Quanterix, Lexington, MA) on the automated Simoa® SR-X analyzer (Quanterix);Brain-derived neurotrophic factor (BDNF) and Neurogranin levels by using commercially available arrays, employing the Bio-Plex 200 System array reader (Bio-Rad, Hercules, CA, USA).

Biological samples will be stored in the BioBank of the IRCCS Istituto Centro San Giovanni di Dio, Fatebenefratelli.

Structural and functional connectivity will be measured using the following: Resting-state functional magnetic resonance imaging (MRI) to derive functional connectivity matrices;Diffusion-weighted MRI to derive structural connectivity matrices.

In detail, a high-field 3T scanner will be used, and an MRI connectivity study on brain networks mediating language will be adopted. Imaging analysis will be conducted by using a combination of neuroimaging software (Statistical Parametric Mapping (SPM12, http://www.fil.ion.ucl.ac.uk/spm/ (accessed on 6 July 2023)); FSL (https://fsl.fmrib.ox.ac.uk/fsl/fslwiki (accessed on 6 July 2023)), GIFT (http://mialab.mrn.org/software/gift/ (accessed on 6 July 2023)), and FreeSurfer (https://surfer.nmr.mgh.harvard.edu/ (accessed on 6 July 2023)).

### 2.7. Data Collection

Demographic characteristics will be collected at the baseline evaluation (T0). Likewise, screening tools for handedness (Edinburgh Handedness Inventory (EHI, [67]), screening checklist for the application of tDCS, and Cognitive Reserve Index questionnaire (CRI-q, [68]) will be administered at the baseline assessment (T0). Data from clinical, cognitive, and language measures (primary and secondary outcomes) will be collected at the baseline (T0) and at each timepoint of evaluation (T1 and T2). MRI and biomolecular data will be collected only at T0 and T1. Primary and secondary outcome measures will be assessed for a single patient by the same trained neuropsychologist (blinded to patient treatment allocations) for all visits.

Moreover, at the end of each tDCS session, all participants will be asked to complete questionnaires to assess any adverse events and sensations induced by the tDCS [69]. After completing the last treatment session, participants will be asked whether they believe they received the active or sham stimulation (dichotomous response).

### 2.8. Statistical Analysis

A repeated-measures analysis will be used to compare pre-treatment (T0) versus post-treatment (T1) and follow-up (T2) variables for the two groups. Regarding MRI, the data will be analyzed using repeated-measures ANOVAs by testing different experimental factors: timing (T0, T1), hemisphere, and electrode site. Correlations between cognitive/functional changes, imaging biomarkers, and molecular biomarkers will be assessed by using Pearson’s correlation coefficient.

## 3. Discussion

The treatment of language difficulties in PPA remains a real challenge and a high-priority unmet medical need. A radical change in the way we approach this disease is necessary; the MAINSTREAM protocol represents an opportunity in this regard, with the aim of developing a novel, non-invasive, and low-cost intervention. This protocol will evaluate the effect of tDCS neuromodulation coupled with individualized language training in a sample of patients with mild PPA. In particular, we aim to establish whether the combined use of active tDCS and individualized language training in mild PPA patients can induce changes in cognitive and functional performance, brain activity, and molecular markers. 

These results could replicate and further validate the available findings regarding the effectiveness of neuromodulation techniques combined with individualized language training [27]. Some studies have suggested that language training, alone or in combination with tDCS, improves oral naming accuracy for trained items in patients with PPA, with long-term maintenance of the gains [12,27,30]. However, only language training combined with tDCS provides a generalization effect, enhancing oral naming accuracy also for untrained items [33,38,70,71]. Nevertheless, little is known about the underlying neural mechanisms of tDCS. 

Understanding the brain mechanisms supporting successful intervention is essential for scientific and clinical reasons. Recent studies have reported modulations in the activated brain network [72,73] as well as in synaptic transmission and plasticity [74,75], indicating that tDCS may work physiologically by altering cell membrane potentials and thus affecting the synaptic conductivity of neurons [76,77,78]. In light of these studies, our protocol has several strengths. First, it represents a fully translational project, and the long-range goal of the proposed line of research is to create clinical protocols directly available to patients. Specifically, this novel approach could make essential advancements in the care of plPPA by creating new rehabilitation instruments for daily disease management that substantially reduce both health expenses and the burden on families. Positive findings of this study would demonstrate a novel, effective, non-invasive and non-pharmacological treatment for PPA patients. Moreover, this protocol will increase the overall understanding of the neural and biological characteristics linked to the effect of language treatment and non-invasive brain stimulation applications in PPA.

## 4. Conclusions

The treatment of PPA can be considered a high-priority unmet medical need. The disorder has a high ratio of underdiagnosis, and no specific epidemiological data is available. Based on the frontotemporal dementia incidence data from Italy, an estimate is 3.05 for 100.000 person-years [79], suggesting a 30 to 40 per cent PPA phenotype for the condition [80]. The potential of this approach is a substantial advancement in the rehabilitation field, and it will yield novel approaches for treating cognitive deficits in patients with PPA. The findings of this study could represent a valuable contribution to the research field, with translational applications in the treatment and monitoring of PPA. Moreover, tDCS is a low-cost and easy-to-use technique, which would thus facilitate its availability in care. The obtained data will help advance the construction of an innovative and more effective therapeutic strategy with potential implications for public health. 

Finally, our project could pave the way for further studies aimed at investigating the efficacy of combined treatments (e.g., the stimulation of other areas during the execution of other tasks) and verifying the usefulness of these programs in other patient populations (e.g., mild cognitive impairment).

## Figures and Tables

**Figure 1 brainsci-13-01060-f001:**
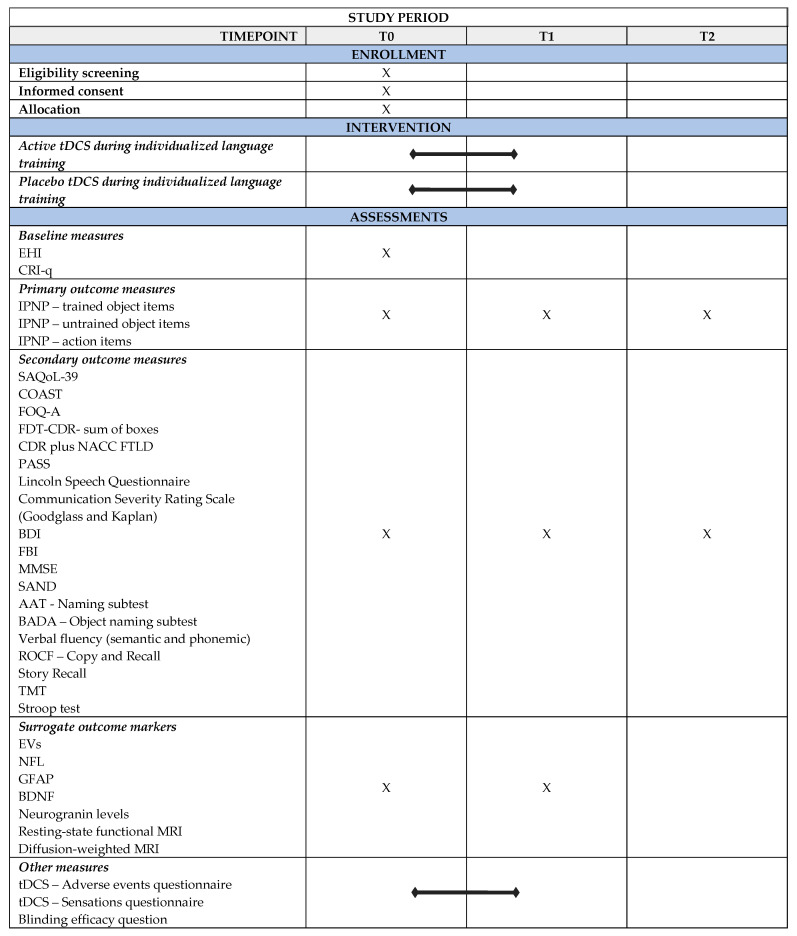
SPIRIT figure for the enrollment, intervention, and assessment schedule in the study. AAT = Aachener Aphasia Test; BADA = Battery for the Assessment of Aphasic Disorders; BDI = Beck Depression Inventory; BDNF = Brain-derived neurotrophic factor; CDR plus NACC FTLD = Global Clinical Dementia Rating plus National Alzheimer’s Coordinating Center Frontotemporal Lobar Degeneration; COAST = Communication Outcome After Stroke; CRI-q = Cognitive Reserve Index-questionnaire; EHI = Edinburgh Handedness Inventory; EVs = Extracellular vesicles; FBI = Frontal Behavioral Inventory; FTD—CDR = Frontotemporal Dementia—Clinical Dementia Rating Scale; FOQ—A = Functional Outcome Questionnaire-aphasia; GFAP = Glial Fibrillary Acidic Protein; IPNP = International Picture Naming Project Task; MMSE = Mini Mental State Examination; MRI = Magnetic Resonance Imaging; NFL = Neurofilament light chain; PASS = Progressive Aphasia Severity Scale; ROCF = Rey—Osterrieth Complex Figure; SAND = Screening for Aphasia in NeuroDegeneration; SAQoL—39 = Stroke and Aphasia Quality of Life Scale—39; tDCS = transcranial Direct Current Stimulation; T0 = Baseline Assessment; T1 = Post-treatment Assessment; T2 = Follow-up Assessment; TMT = Trail Making Test.

**Figure 2 brainsci-13-01060-f002:**
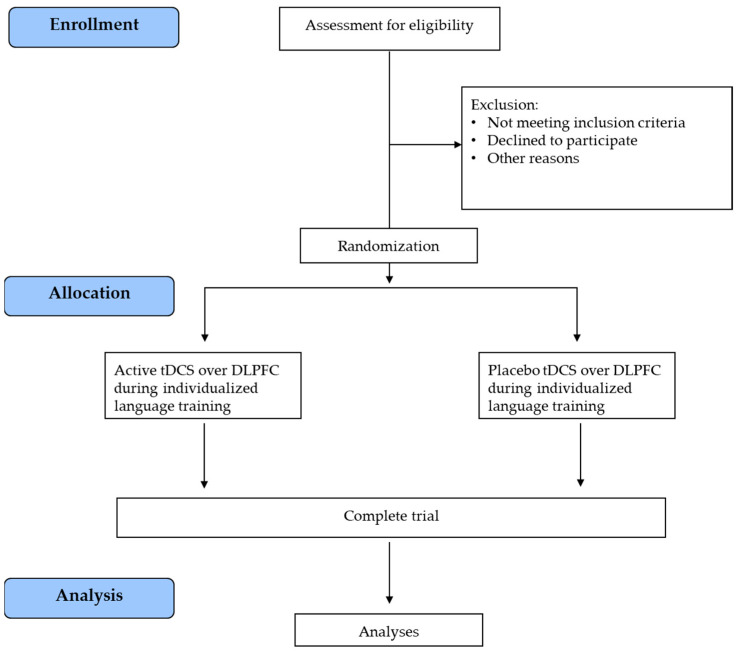
Trial work plan. tDCS = transcranial Direct Current Stimulation; DLPFC = dorsolateral prefrontal cortex.

## Data Availability

Data sharing is not applicable to this article.

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
