# Peer review of "A Multimodal Approach for Clinical Diagnosis and Treatment of Primary Progressive Aphasia (MAINSTREAM): A Study Protocol"

_brainsci, 2023, doi:10.3390/brainsci13071060_

Round 1

Reviewer 1 Report

Dear Authors,

I read your work entitled “A Multimodal Approach for Clinical Diagnosis and Treatment of Primary Progressive Aphasia (MAINSTREAM): a study protocol” and here I enclose my recommendations to you:

1.     The “Introduction” section is written very well and provides a clear rational.

2.     The “Methods” section is written in a good manner, but sure extra information’s are needed. I suggest the Authors to add more info about the participants if all scales are standardised in their languages and population. If not, the authors must provide rational why these scales were selected even there not standardized. Τhere is a need for a more detailed description for “2.5.2. Language training” with an adequate number of references.

Thank you.

Author Response

RW POINT 2.The “Methods” section is written in a good manner, but sure extra information’s are needed. I suggest the Authors to add more info about the participants if all scales are standardised in their languages and population. If not, the authors must provide rational why these scales were selected even there not standardized. Τhere is a need for a more detailed description for “2.5.2. Language training” with an adequate number of references.

RESPONSE: We thank the reviewer for this point. We have now specified that scales and neuropsychological tests are standardized in Italian language. In the protocol study we also added two functional scales (Lincoln Speech Questionnaire and Communication Severity Rating Scale). In both cases, Italian translations are available, but they have not been standardized.  As such, these scales are applied only for recollect additional clinical information. Finally, we have included more detail of treatment in the paragraph ‘Language training’.

Reviewer 2 Report

This paper can have its merits. 

However, its English language is evidently non-native and, despite it is clear, needs a thorough revision, especially at the level of style, with the help of a native speaker, which would be truly beneficial. 

The issues, at the level of language and written style, start already from the beginning of the paper (even from the Abstract). Look at the first sentence, for instance, "Spreading the results of [...]". "Spreading"? Do the Authors mean "Sharing"? And so on, and so forth. A lot of lexical repetitions plague the text - they should be fixed and removed (synonyms exist for a reason). 

The Introduction is ok. 

But where is the Literature Review? 'Scattered' here and there, over the paper, starting from the Introduction. Why not to implement a proper section, entitled "Literature Review", collecting and analyzing all the studies used and cited in the paper, with the addition of some more general works, which will make the article more 'user-friendly', even for a non-specialized readership? 

The methodology is ok, and looks solid. However, it should be double-checked by a Reviewer with a more direct specialization than me in the topic, not only inherently in reproducibility, but also for final validation. 

The Discussion is extremely poor. It should be the 'meat' of the paper, but it is just a final comment which could be a Conclusion, but not a Discussion - part of the possible Discussion is included into the previous sections, which are mostly Results, in any case. 

In light of this, the paper would need a deep re-working of its format, to make it 'tidy' and, especially, up to academic standards. 

And where is the Conclusion? A scientific paper without Conclusion is quite awkward. The Discussion needs to be enhanced and expanded - considerably, by adding more analysis and comments. Additionally, an independent Conclusion should be implemented, with a summary of the research goals of the paper and with a (further) stress on how the Authors have achieved them. 

Moreover, both in the Introduction and in the Conclusion, like in a 'mirror', the Authors should state and/or explain why this paper is relevant in its panorama of studies, and highlight more what it adds to our knowledge and, ultimately, to Science in general. 

All in all, the article is promising and the project has its merits, but the text is written not very well, the structure is lacking (of) many important elements, and more reflections and observations are necessary to make it strong enough to be considered for publication. 

Thank you very much. 

As mentioned, the English language of the paper is evidently non-native and, despite it is clear, needs a thorough revision, especially at the level of style, with the help of a native speaker, which would be truly beneficial. 

The issues, at the level of language and written style, start already from the beginning of the paper (even from the Abstract). Look at the first sentence, for instance, "Spreading the results of [...]". "Spreading"? Do the Authors mean "Sharing"? And so on, and so forth. A lot of lexical repetitions plague the text - they should be fixed and removed (synonyms exist for a reason). 

Author Response

REV2

This paper can have its merits. 

Response: We thank the Reviewer for his/her time and efforts necessary to review the manuscript. We sincerely appreciate all valuable comments and suggestions.

However, its English language is evidently non-native and, despite it is clear, needs a thorough revision, especially at the level of style, with the help of a native speaker, which would be truly beneficial. The issues, at the level of language and written style, start already from the beginning of the paper (even from the Abstract). Look at the first sentence, for instance, "Spreading the results of [...]". "Spreading"? Do the Authors mean "Sharing"? And so on, and so forth. A lot of lexical repetitions plague the text - they should be fixed and removed (synonyms exist for a reason). The Introduction is ok. But where is the Literature Review? 'Scattered' here and there, over the paper, starting from the Introduction. Why not to implement a proper section, entitled "Literature Review", collecting and analyzing all the studies used and cited in the paper, with the addition of some more general works, which will make the article more 'user-friendly', even for a non-specialized readership? 

Response: We agree with the Reviewer that the English language could be improved. As such, we have employed the help of a native English speaker to revise the entire manuscript. Moreover, following the Reviewer’s suggestion, we have now added the ‘Prior research (Literature review)’ paragraph.

The methodology is ok, and looks solid. However, it should be double-checked by a Reviewer with a more direct specialization than me in the topic, not only inherently in reproducibility, but also for final validation.

The Discussion is extremely poor. It should be the 'meat' of the paper, but it is just a final comment which could be a Conclusion, but not a Discussion - part of the possible Discussion is included into the previous sections, which are mostly Results, in any case. In light of this, the paper would need a deep re-working of its format, to make it 'tidy' and, especially, up to academic standards. And where is the Conclusion? A scientific paper without Conclusion is quite awkward. The Discussion needs to be enhanced and expanded - considerably, by adding more analysis and comments. Additionally, an independent Conclusion should be implemented, with a summary of the research goals of the paper and with a (further) stress on how the Authors have achieved them. Moreover, both in the Introduction and in the Conclusion, like in a 'mirror', the Authors should state and/or explain why this paper is relevant in its panorama of studies, and highlight more what it adds to our knowledge and, ultimately, to Science in general.

Response: We are grateful to the Reviewer for calling our attention to this issue. We have now extensively revised the Discussion section along with having added a separate section for the Conclusion.

All in all, the article is promising and the project has its merits, but the text is written not very well, the structure is lacking (of) many important elements, and more reflections and observations are necessary to make it strong enough to be considered for publication. 

Thank you very much. 

Comments on the Quality of English Language

As mentioned, the English language of the paper is evidently non-native and, despite it is clear, needs a thorough revision, especially at the level of style, with the help of a native speaker, which would be truly beneficial. 

The issues, at the level of language and written style, start already from the beginning of the paper (even from the Abstract). Look at the first sentence, for instance, "Spreading the results of [...]". "Spreading"? Do the Authors mean "Sharing"? And so on, and so forth. A lot of lexical repetitions plague the text - they should be fixed and removed (synonyms exist for a reason).

Response: we have employed the help of a native English speaker to revise the entire manuscript.

Round 2

Reviewer 2 Report

The paper has been improved. 

  It can be considered, now, for publication. 

  Perhaps, it could be expanded still a little, at the level of comments and analysis. 

  The English language, here and there, still requires some care. 

  Thank you very much. 

The English language still requires some care. 

  Look, among others, at this sentence, "Treating language difficulties in PPA remains a real challenge and a high-priority unmet medical need." 

  Possibly, the MDPI professionals can take care of this. 

  Some inconsistencies, also, should be fixed. 

  All in all, it was improved, in any case.